# Adaptive Memory Mechanism in Vision Transformer for Long-form Video Understanding

## Abstract

In long-form video understanding, selecting an optimal Temporal Receptive Field (TRF) is crucial for Vision Transformer (ViT) models due to the dynamic nature of diverse video motion contents, which varies in duration and velocity. A short TRF can result in loss of critical information, while a long TRF may decrease ViT's performance and computational efficiency caused by the unrelated contents in videos and the quadratic complexity of the attention mechanism. To tackle this issue, we introduce Adaptive Memory Mechanism (AMM) that enables ViT to adjust its TRF dynamically in response to the video's dynamic contents. Instead of discarding Key-Value (KV) cache from the earliest inference when the settings limit is reached, our approach uses a Memory Bank (MB) to retain the most important embeddings from the Key-Value cache that would otherwise be discarded in memory-augmented methods. The selection is based on the attention score calculated between the Class Token (CLS) in current iteration and the KV cache in previous iterations. We demonstrate that Adaptive Memory Vision Transformer (AMViT) outperforms existing methods across a diverse array of tasks (action recognition, action anticipation, and action detection).

## 1 Introduction

The Vision Transformer (ViT) has recently been found effective for diverse computer vision tasks. By leveraging the self-attention mechanism to capture long-range dependencies across image patches, it can outperform convolutional neural networks (CNNs) and achieve the state-of-the-art performance in both image (Russakovsky et al., 2015) and video (Kay et al., 2017) (Carreira et al., 2018) (Carreira et al., 2022) domains. Notwithstanding its success, ViT's application to video understanding is constrained by its inherent computational design, particularly when processing long-form videos. The quadratic complexity of the self-attention mechanism creates a computational bottleneck, limiting the temporal receptive field (TRF) that can be supported and, consequently, the model's ability to integrate information over extended time periods.

To mitigate the computational demands so as to extend the TRF without incurring prohibitive computation costs, memory-augmented methods such as MeMViT (Wu et al., 2022) have been introduced. These methods incorporate mechanisms like stop-gradient Key-Value (KV) cache that draw parallels to truncated backpropagation-through-time (BPTT) (Mikolov et al., 2010), enabling the extension of the TRF without much increase in computation. Subsequent work (Balazevic et al., 2024) has introduced compression methods that do not require training. Most memory-augmented methods manages the KV cache using strategies such as First-In-First-Out (FIFO), which may lead to the discarding of salient information when newer, but less relevant, KV pairs displace older, more pertinent ones. It is common for a video to contain contents of different actions which may last for different periods of time. For instance, as shown in Figure 1, actions like "wash pot" typically last for a long period of time than actions like "take pot" in a kitchen scene. The former one needs a longer TRF, while the latter needs only a short one.

To this end, drawing inspiration from some recent works (Oren et al., 2024; Ge et al., 2024) which showed that carefully selecting KV pairs for caching is effective for Large Language Models (LLMs), we propose an Adaptive Memory Mechanism (AMM) to enable the ViT to adaptively prioritize the retention of embeddings based on their attention score with the Class Token (CLS) in the current iteration. The retained embeddings will be stored in a Memory Bank (MB) which is then

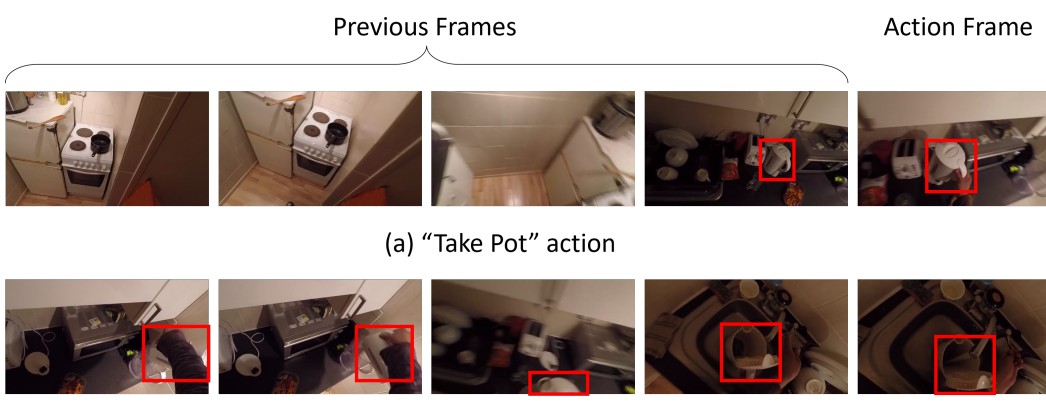

(a) "Take Pot" action

(b) "Wash Pot" action

Figure 1: Illustration of the need of adaptive temporal receptive fields in ViT for understanding different actions in a video. The frames above are extracted from a video at 1 frame per second. (a) The"Take Pot" action frame only has one preceding frame associated with the "Pot" (indicated using a red bounding box). (b) The "Wash Pot" action instead lasts for four frames before the action frame, and the corresponding temporal receptive field should be adapted accordingly in an input-aware manner.

used to generate memory-augmented Keys and Values for ViT. By leveraging the Class Token for updating the Memory Bank iteratively, the augmented Keys and Values own adaptive TRF according to the time-varying video content. We call the ViT with AMM incorporated *Adaptive Memory Vision Transformer* (AMViT), which not only mitigates the constraints of fixed TRF but also enhances the ViT's capability to discern and utilize pertinent temporal information.

We conduct a comprehensive performance evaluation of the proposed AMViT and demonstrate its superior performance over the vanilla ViT and the SOTA memory-augment model MeMViT (Wu et al., 2022) using two video datasets, including AVA (Gu et al., 2018) and Epic-Kitchens (Damen et al., 2018). Our results indicate that AMViT not only achieves higher accuracy in understanding long-form video content, but also does so with reduced computation overhead during training, thereby addressing both the effectiveness and efficiency concerns that currently limit the application of ViT models in understanding long-form video.

## 2 RELATED WORK

**Vision Transformers**    The advent of Transformer architectures (Vaswani et al., 2017), initially designed for NLP tasks, revolutionized multiple domains due to their remarkable performance. In the realm of Computer Vision (CV), Vision Transformers (ViTs) have emerged as a powerful alternative to CNNs. Pioneering this shift, Dosovitskiy et al. (2021) introduced a method to process images by dividing them into non-overlapping patches and treating them as embeddings, analogous to tokens in NLP. This approach has been extended to video tasks, where SpatioTemporal Attention (STA) (Bertasius et al., 2021) is utilized to handle patches from consecutive frames, as seen in works such as Tong et al. (2022) and Li et al. (2023).

**Memory-augmented Methods**    In NLP, Transformer-XL (Dai et al., 2019) introduced the concept of Key-Value (KV) cache as Memory, enabling the model to retain and leverage information from the KV cache. This method has shown significant improvements for tasks requiring the comprehension of extended textual sequences. Subsequent enhancements include the development of Compressive Memory method (Rae et al., 2020), which aims to condense the Memory, and the use of linear attention's KV cache combined with the Deta Rule (Schlag et al., 2021) as more efficient Memory from Munkhdalai et al. (2024). Transferring these NLP-inspired memory methods to CV, works like Wu et al. (2022) have adapted the Compressive Memory method for ViT, facilitating

the understanding of long-form video. Despite these advancements, methods which could adapt the temporal receptive field (TRF) based on the dynamic video content are still lacking.

**Long-form Video Understanding**    The challenge of long-form video understanding has been approached by various methods trying to extend the TRF. Some methods (Wang et al., 2023) (Xu et al., 2021) (Rodriguez-Opazo et al., 2023) scale the TRF to accommodate the entire video sequence in one pass. For instance, Wang et al. (2023) employs a Gumbel softmax mechanism (Jang et al., 2017) to identify and retain salient features, while Xu et al. (2021) proposes a two-stage encoder to distill long video sequences into concise representations suitable for ViT processing. Rodriguez-Opazo et al. (2023) suggests the use of I3D features as inputs, which allows for a more compact feature representation and an expanded content capacity for ViT. Alternative methods, such as those proposed in Wu et al. (2022) and Balazevic et al. (2024), utilize past KV cache as Memory, enabling multiple passes over sequentially video segments and allowing the model to obtain information from previous video segments. Our method builds upon the memory-augmented methods, employing novel Adaptive Memory Mechanisms to enhance the understanding of long-form videos.

## 3    ADAPTIVE MEMORY VISION TRANSFORMER

In this section, we first present the vanilla Vision Transformer and its memory-augmented variants with KV caching incorporated. We then introduce the proposed Adaptive Memory Mechanism and explain how it can enable dynamic temporal receptive field for KV caching.

### 3.1    VISION TRANSFORMER

In video understanding, existing Vision Transformer (ViT) methods (Tong et al., 2022) (Li et al., 2023) employ SpatioTemporal Attention (STA) (Bertasius et al., 2021) to facilitate interaction among all tokens within the video sequence. Each video frame is first partitioned into $n$ non-overlapping patches are then projected into embeddings. The embeddings of consecutive $t$ frames are then reorganized into a matrix $\boldsymbol{X}' \in \mathbb{R}^{t \cdot n \times d}$ , where $t$ is the number of frames, $n$ is the number of embeddings (patches) per frame, $t \cdot n$ gives the total number of embeddings, and $d$ is the dimensionality of the embeddings.

ViT also includes in its input a Class Token (CLS) which is a token that represents the class label of the entire input sequence in the form of a corresponding embedding $\check{x}_{CLS}$. The input thus becomes $\boldsymbol{X} \in \mathbb{R}^{(t \cdot n + 1) \times d}$

$$\boldsymbol{X} = [\check{x}_{CLS}, \boldsymbol{X}'] \tag{1}$$

where the square brackets denote concatenation along the token dimension. The input then passes through a stack of transformer layers, each with an Attention Operation (Vaswani et al., 2017) which linearly projects the input matrix $\boldsymbol{X}$ to be query $\boldsymbol{Q}$, key $\boldsymbol{K}$, and values $\boldsymbol{V}$:

$$\boldsymbol{Q} = \boldsymbol{X}\boldsymbol{W_Q}; \boldsymbol{K} = \boldsymbol{X}\boldsymbol{W_K}; \boldsymbol{V} = \boldsymbol{X}\boldsymbol{W_V} \tag{2}$$

and compute $\boldsymbol{Z} \in \mathbb{R}^{(t \cdot n + 1) \times d}$

$$\boldsymbol{Z} = \mathrm{Attn}(\boldsymbol{Q}, \boldsymbol{K}, \boldsymbol{V}) \tag{3}$$

to be fed to a Multilayer Perceptron (MLP) to obtain the output $\boldsymbol{O} \in \mathbb{R}^{(t \cdot n + 1) \times d}$. The output then forms the input for the next transformer layer.

Vanilla ViT essentially takes the $t$ preceding video frames as its temporal receptive field (TRF). While we can increase the value $t$ to allow longer dependency to be captured in long-form videos, the computation complexity also increases quadratically.

### 3.2    MEMORY AUGMENTED VISION TRANSFORMER

With the objective to extend the TRF in a cost-effective manner, memory-augmented methods (Rae et al., 2020; Wu et al., 2022) split a video into a sequence of short segments of $t$ frames and process them sequentially (for both training and inference). Memory is used to store at each processing iteration some representations of the processed segments which will be used in the next processing iteration. When processing the video segment at iteration $i$, Memory-augmented ViT makes access

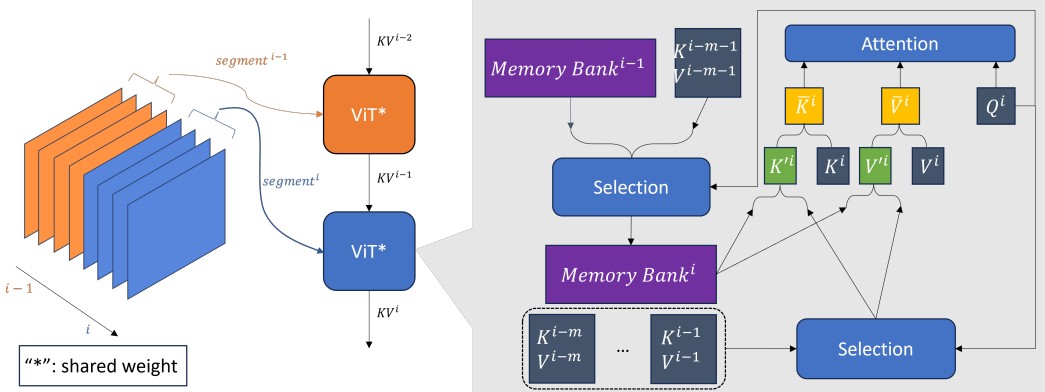

Figure 2: An overview of the proposed adaptive memory mechanism (AMM). The lefthand side shows how a long-form video is fed to a temporally unfolded Vision Transformer (ViT). The right-hand side shows how the proposed AMM is introduced in the attention layer. In iteration $i$, before dropping $KV^{i-m-1}$ cache, embeddings in $MemoryBank^{i-1}$ and $KV^{i-m-1}$ are first selected with reference to the Class Token from $Q^i$ and then aggregated to obtain $MemoryBank^i$. Some longer-range but useful information can thus be retained in the Memory Bank. $MemoryBank^i$ is further concatenated with embeddings selected from $KV^{i-1}$ back to $KV^{i-m}$ to obtain $K'V'^i$. The selective caching from $KV^{i-1}$ back to $KV^{i-m}$ further helps the ViT reduce redundancy to consider only useful tokens.

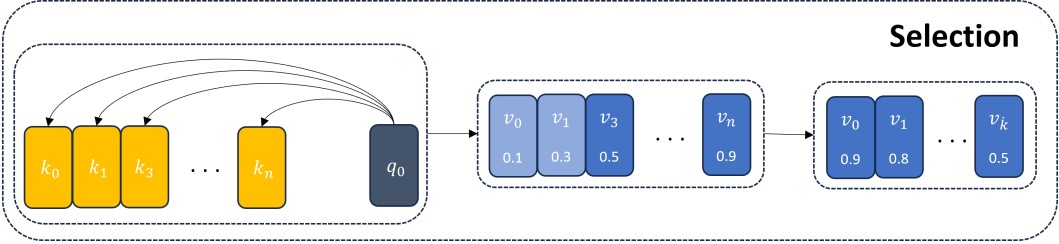

Figure 3: Procedures of Values selection. Class token $q_0$ from Querys inner product with the Keys to get the attention score. Top-$\dot{k}$ Values embeddings with highest attention score are the output of the selection.

to not only the current segment but also the information of previous segments in earlier iterations $i' < i$ stored in the Memory. In particular, it is common to store the Key ($K$) and Value ($V$) from the previous iterations in Memory, and concatenate $K^i$ and $V^i$ in the current iteration $i$ with $K^{i'}$ and $V^{i'}$ cached from the earlier iterations $i'$, where $i'$ ranges from $i-m$ to $i-1$, that is,

$$\overline{K}^i = [\, cpr(K_{sg}^{i-m}), \, ..., \, cpr(K_{sg}^{i-1}), \, K^i \,] \tag{4}$$

$$\overline{V}^i = [\, cpr(V_{sg}^{i-m}), \, ..., \, cpr(V_{sg}^{i-1}), \, V^i \,] \tag{5}$$

$$Z = \text{Attn}(Q^i, \overline{K}^i, \overline{V}^i) \tag{6}$$

where the square bracket denotes concatenation along the token dimension, "$sg$" represents stop gradient (flush the gradient from the previous iterations), "$cpr$" represents the Convolution Layer used to compress the Memory, and $m$ is the memory length. With this concatenation, the memory-augmented $\overline{K}^i$ and $\overline{V}^i$ contain not only the information from the current iteration $i$, but also the information from up to $m$ iterations before.

### 3.3 ADAPTIVE MEMORY MECHANISM

Memory-augmented methods (Rae et al., 2020; Wu et al., 2022) aim to extend the temporal receptive field (TRF) of the ViT architecture by a factor of m. While these approaches have shown promising results, the fixed TRF imposed by the first-in-first-out (FIFO) mechanism does not align well with the dynamic nature of video content, as illustrated in Figure 1, and the memory-augmented Keys and Values still face the redundancy issues mentioned in Wu et al. (2022). To address the limitations from FIFO mechanism, we propose Adaptive Memory Mechanism (AMM) that utilize Memory Bank (MB) to retain embeddings releted to the input in current iteration by Input-aware Selective Module (ISM). The AMM allows ViT to achieve a more flexible TRF that responds dynamically to the videos.

**Input-aware Selective Module (ISM)**   MeMViT uses learnable compression modules to reduce the spatiotemporal size of the $KV$ caches in memory. However, it is agnostic to the input of the current iteration. To allow embedding relevant to the input in current iteration to be retained, we first propose an Input-aware Selective Module (ISM). Specifically, we select the top-$\dot{k}$ most relevant embeddings with reference to the Class Token (CLS). Formally, we define:

$$selection(\boldsymbol{K}, \boldsymbol{q}_0, \dot{k}) = \boldsymbol{K}_{\mathcal{I}}, \quad \text{where } \mathcal{I} = \arg\max_{J \subset \Delta_{|\boldsymbol{K}|}, |J|=\dot{k}} \sum_{j \in J} \boldsymbol{k}_j^\top \boldsymbol{q}_0 \tag{7}$$

where $|\boldsymbol{K}|$ is the embedding number of $\boldsymbol{K}$, $\Delta_{|\boldsymbol{K}|}$ denotes all subsets of the set $\{1, 2, .., |\boldsymbol{K}|\}$, $\mathcal{I}$ is the index set of the $M$ tokens in $\boldsymbol{K}$ that give the highest values of the inner product between the Key embedding $k_j$ and the class token $\boldsymbol{q}_0$.

**Memory Bank (MB)**   As illustrated previously, understanding different actions in videos requires adaptive temporal receptive fields and the important information may be beyond the fixed memory size $m$. When the $KV$ caches reach its length limit, MeMViT discards the earliest memory entirely. This may result in information essential to video understanding being removed. To fully capture the necessary information beyond the memory limit, we construct a memory bank that is updated in each iteration, given by:

$$MB_k^i = \left[ selection(cpr(\boldsymbol{K}_{sg}^{i-m-1}), \boldsymbol{q}_0^i, L - \dot{k}'), selection(MB_k^{i-1}, \boldsymbol{q}_0^i, \dot{k}') \right], \tag{8}$$

$$MB_v^i = \left[ selection(cpr(\boldsymbol{V}_{sg}^{i-m-1}), \boldsymbol{q}_0^i, L - \dot{k}'), selection(MB_v^{i-1}, \boldsymbol{q}_0^i, \dot{k}') \right], \tag{9}$$

where $L$ is the size of the Memory Bank, $\dot{k}' = \lfloor \alpha L \rfloor$, $L - \dot{k}'$ is the number of tokens to be selected from the earliest $KV$ cache that is about to discard, $\alpha$ controls the ratio of tokens to be retained from Memory Bank in $i - 1$ iteration.

**Adaptive Memory Mechanism (AMM)**   Then, we can augment the Keys and Values in each iteration with the adaptive memory by replacing Eq. (4) and Eq. (5) with the followings:

$$\overline{\boldsymbol{K}}^i = \left[ MB_k^i, selection(cpr(\boldsymbol{K}_{sg}^{i-m}), \boldsymbol{q}_0^i, \dot{k}''), ..., selection(cpr(\boldsymbol{K}_{sg}^{i-1}), \boldsymbol{q}_0^i, \dot{k}''), \boldsymbol{K}^i \right] \tag{10}$$

$$\overline{\boldsymbol{V}}^i = \left[ MB_v^i, selection(cpr(\boldsymbol{V}_{sg}^{i-m}), \boldsymbol{q}_0^i, \dot{k}''), ..., selection(cpr(\boldsymbol{V}_{sg}^{i-1}), \boldsymbol{q}_0^i, \dot{k}''), \boldsymbol{V}^i \right] \tag{11}$$

The overall effect is that the iterative updating of the Memory Bank tries to retain useful embeddings from the dropped $KV$ caches and the $selection$ function tries to prevent retain redundant embeddings in the Memory Bank of ViT as far as possible.

## 4 EXPERIMENT

To evaluate the effectiveness of AMViT (ViT with AMM incorporated), two datasets with long-form videos are used for benchmarking.

### 4.1 DATASET

**AVA dataset** (Gu et al., 2018) was created to benchmark the ability of models to localize and identify human actions in extended video sequences. The dataset comprises over 430 15-minute video clips extracted from movies and television shows, resulting in a diverse representation of scenes and scenarios. Each video in AVA is densely annotated with action labels at one-second intervals, with an average of 1 to 12 labels per person per second, spanning 80 different action categories.

**Epic-Kitchens dataset** (Damen et al., 2018) is a first-person (egocentric) video dataset. It encompasses over 55 hours of video data, recorded by 32 participants across multiple kitchen environments. The dataset is annotated with a rich vocabulary of object interactions and actions, featuring over 39,000 action segments and 3,806 unique action classes. Each action is defined by a verb-noun pair, providing a detailed account of the interaction dynamics inherent to cooking activities.

### 4.2 IMPLEMENTATION DETAILS

**Data Loading**    To implement the memory-augmented methods, both during the training and inference phases, we adopt an online processing approach for the video data, which involves sequential reading of consecutive segments. This aligns well with real-world applications such as robotic vision systems and live video streaming understanding, where data is inherently processed in a sequential manner. To facilitate this, following Wu et al. (2022), we concatenate the entirety of our video dataset and process it as a continuous stream. For the episode at the boundary of two consecutive segments from different videos where there is context discontinuity, we adopt a masking strategy that initializes the AMM to ensure that the memory from a previous video does not influence the processing of a subsequent one.

**Backbone Selection**    We adopt the pre-trained UMT's ViT as the backbone in this paper, as underpinned by the findings of Amos et al. (2024) which suggests pre-training, even with a focus on short TRF relationship, can be beneficial for understanding long TRF relationship in data. By utilizing a pre-trained ViT, we could capitalize on the pre-existing learned representations and substantially reduce the computational overhead that would be associated with training ViTs from scratch. The backbones were pre-trained on the K710 dataset, a comprehensive fusion of K400 (Kay et al., 2017), K600 (Carreira et al., 2018), and K700 (Carreira et al., 2022) datasets, as described in Li et al. (2023).

**Model Setting**    To ensure fair performance comparison with the state-of-the-art models including UMT (Li et al., 2023) and MeMViT Wu et al. (2022), we try to standardize the backbone architecture and the training hyperparameters for both models as well as the proposed AMViT. Thus, we change the MeMViT's backbone from MViT to UMT (see Appendix A.2 for details).

**Positional Embedding**    Following UMT (Li et al., 2023), we utilize absolute positional embedding (Vaswani et al., 2017) so as to be compatible with the UMT's pre-trained model and to benefit from the reduced memory requirement.

### 4.3 OVERALL PERFORMANCE

We evaluate the performance of ViT (no memory-augmented), MeMViT, and AMViT across three tasks: action detection, action recognition, and action anticipation. Our analysis includes two variants of the backbone model, ViT-B and ViT-L, both pretrained on the K710 dataset.

#### 4.3.1 ACTION DECTION IN AVA DATASET

The action detection task within the AVA dataset (Gu et al., 2018) presents significant challenges, as it necessitates the simultaneous localization and recognition of actions across 80 categories. To evaluate model performance, we employ the mean Average Precision (mAP) metric. Building on the work of UMT (Li et al., 2023), we utilize the object detection files provided by Tang et al. (2020) to bypass the need for person detection during validation.

Table 1 presents the performance for the action detection task in the AVA dataset. For the base variant (ViT-B, MeMViT-B, and AMViT-B), AMViT demonstrates superior accuracy with an mAP of

Table 1: Performance for action detection in the AVA dataset.

| Model | Pre-trained | FLOPs(G) | Param(M) | mAP(%) |
|---|---|---|---|---|
| ViT-B | K710 | 202 | 85.7 | 28.59 |
| MeMViT-B | K710 | 202 | 85.7 | 29.17 |
| **AMViT-B** | K710 | 202 | 85.7 | **30.07** |
| ViT-L | K710 | 714 | 307 | 34.44 |
| MeMViT-L | K710 | 714 | 307 | 35.91 |
| **AMViT-L** | K710 | 714 | 307 | **35.98** |

30.07%, which is a significant improvement over the MeMViT's 29.17% and the baseline ViT's 28.59%. Similarly, in the large variant models (ViT-L, MeMViT-L, and AMViT-L), AMViT-L achieves an mAP of 35.98%, marginally outperforming MeMViT-L's 35.91% and significantly surpassing ViT-L's 34.44%. It is noteworthy that this enhancement in performance does not much come at the cost of increased parameters and FLOPs. The results indicate that AMViT is capable of capturing more nuanced features relevant to the granular action categories within the AVA dataset. The improvement in mAP suggests that AMVIT can more accurately localize and classify multiple actions within a given frame.

### 4.3.2 ACTION RECOGNITION AND DETECTION IN EPIC-KITCHENS DATASET

We further evaluate the proposed model using the Epic-Kitchens dataset with the action recognition and action anticipation tasks. These tasks are particularly challenging due to the fine-grained and diverse nature of kitchen activities and the first-person viewpoint from which the videos are recorded.

Table 2: Performance for action recognition in the EPIC-KITCHEN-100 dataset.

| Model | Pre-trained | FLOPs(G) | Param(M) | Top-1 Accuracy(%) |
|---|---|---|---|---|
| ViT-B | K710 | 202.22 | 85.64 | 37.12 |
| MeMViT-B | K710 | 202.28 | 85.66 | 37.87 |
| **AMViT-B** | K710 | 202.28 | 85.66 | **38.45** |
| ViT-L | K710 | 714.44 | 306.98 | 42.47 |
| MeMViT-L | K710 | 714.62 | 307.01 | 43.03 |
| **AMViT-L** | K710 | 714.62 | 307.01 | **43.15** |

Table 3: Performance for action anticipation in the EPIC-KITCHEN-100 dataset.

| Model | Pre-trained | FLOPs(G) | Param(M) | Top-1 Accuracy(%) |
|---|---|---|---|---|
| ViT-B | K710 | 202.22 | 85.64 | 19.44 |
| MeMViT-B | K710 | 202.28 | 85.66 | 19.15 |
| **AMViT-B** | K710 | 202.28 | 85.66 | **19.54** |
| ViT-L | K710 | 714.44 | 306.98 | 22.54 |
| MeMViT-L | K710 | 714.62 | 307.01 | 22.04 |
| **AMViT-L** | K710 | 714.62 | 307.01 | **22.63** |

Table 2 presents the performance of the Action Recognition task within the Epic-Kitchens dataset. The baseline ViT-B model obtains the top-1 accuracy of 37.12%, and the MeMViT-B shows a slight improvement with an accuracy of 37.87%. Notably, our AMViT-B model surpasses both the baseline and MeMViT-B models, achieving the best top-1 accuracy of 38.45%. This shows that the proposed adaptive memory mechanism is effective in recognizing complex and diverse actions. We observe similar patterns for models with a larger backbone, where ViT-L and MeMViT-L obtain the top-1 accuracy of 42.47% and 43.03%. Respectively, AMViT-L further improves the top-1 accuracy to

43.15%. This demonstrates that the proposed adaptive memory mechanism benefits the long-form video understanding for both base and large backbone models.

Table 3 summarizes the performance of these models in the Action Anticipation task. ViT-B obtains the top-1 accuracy of 19.44%. The MeMViT-B model shows a slight decrease in performance with an accuracy of 19.15%. In contrast, AMViT-B demonstrates its superiority, achieving a top-1 accuracy of 19.54%. Similarly, for the larger backbone, ViT-L achieves the top-1 accuracy of 22.54% while MeMViT-L shows a slight decline in performance with an accuracy of 22.04%. AMViT-L obtains the top-1 accuracy of 22.63%, underscoring its effectiveness in leveraging memory mechanisms for improved action prediction.

Overall, the results highlight the effectiveness of our AMViT models in action recognition and anticipation tasks in the Epic-Kitchens dataset. The inclusion of adaptive memory mechanisms enhances model performance, illustrating the critical role of temporal context in understanding and predicting actions in complex environments. The performance gains observed across both ViT-B and ViT-L models further emphasize the robustness of our approach at various scales.

## 4.4 PERCENTAGE OF TOKENS IN MEMORY BANK

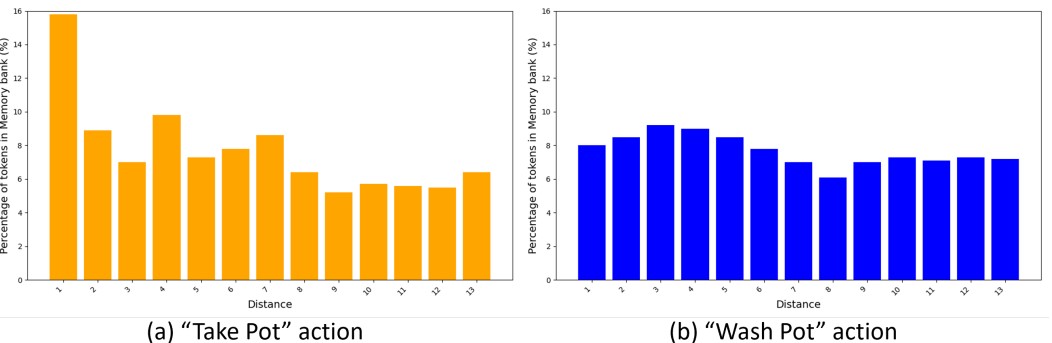

(a) "Take Pot" action          (b) "Wash Pot" action

Figure 4: Memory bank distribution for different actions. The distance indicates the proximity of tokens from previous iterations to the current one. For the "Take Pot" action, the memory bank predominantly retains tokens from recent iterations. In contrast, the "Wash Pot" action shows no distinct preference for tokens from any particular range of past iterations.

Figure 4 illustrates that AMViT consistently maintains a higher percentage of tokens in the vicinity of current iterations for "Take Pot" action than the "Wash Pot" action. This suggests that the Memory Bank puts greater emphasis on recent iterations when inferring the "Take Pot" action. This observation aligns with Figure 1, which shows that the duration of the pot's appearance during the "Take Pot" action is shorter than that in the "Wash Pot" action.

## 4.5 ABLATION EXPERIMENTS

We conduct ablation studies to see how the performance of AMViT is affected by (a) memory bank size, (b) memory length, and (c) the number of memory-augmented layers. All the results are conducted on the Epic-Kitchen (Damen et al., 2018) dataset in action recognition task, and shown in Table 4. The default memory bank size, memory length and memory-augmented layer settings are 50, 2, and 100%.

**Memory Bank Size** ($L$)  Table 4(a) provides insights into the impact of memory bank size on model performance. It is evident that increasing the memory bank size generally enhances the model's accuracy. Specifically, AMViT-B achieves an accuracy of 38.20% with a memory bank size of 40, which improves to 38.45% with a length of 50, and further to 38.49% with a size of 60. This trend suggests that a longer memory bank allows the model to retain more information, thereby boosting its performance.

**Memory Length** ($m$)    Table 4(b) explores the effect of memory length on model performance. MeMViT-B with a memory length of 1 achieves an accuracy of 37.96%, which slightly decreases to 37.87% when the memory length is increased to 2. In contrast, AMViT-B shows an improvement, achieving 38.18% with a memory length of 1 and further increasing to 38.45% with a length of 2. This indicates that while MeMViT experiences a decline in performance with longer memory lengths, AMViT benefits from extended memory, likely due to more effective redundancy reduction.

**Memory-augmented Layers**    Table 4(c) examines the influence of memory-augmented layers on model performance. For MeMViT-B, augmenting 50% of the layers uniformly over the transformer layers results in an accuracy of 37.71%, which slightly improves to 37.87% when all layers are augmented. On the other hand, AMViT-B achieves 38.06% accuracy with 50% uniform augmentation, which further increases to 38.45% when all layers are augmented. This suggests that AMViT-B benefits more significantly from fully augmented layers compared to MeMViT-B, likely due to a more effective integration of augmented information throughout the network.

Table 4: Ablation studies with respect to (a) memory bank size, (b) memory length, (c) memory-augmented layers. (Dataset: Epic-Kitchen; Task: action recognition). "(uni)" means that the augmented layers are uniformly added over the transformer layers of the model.

(a) Memory bank size ($L$)

| Model | $L$ | Acc(%) |
|---|---|---|
| **AMViT-B** | 40 | 38.20 |
| **AMViT-B** | 50 | 38.45 |
| **AMViT-B** | 60 | **38.49** |

(b) Memory length ($m$)

| Model | $m$ | Acc(%) |
|---|---|---|
| MeMViT-B | 1 | 37.96 |
| MeMViT-B | 2 | 37.87 |
| **AMViT-B** | 1 | 38.18 |
| **AMViT-B** | 2 | **38.45** |

(c) Memory-augmented layers

| Model | % of augmented layers | # of augmented tokens | Acc(%) |
|---|---|---|---|
| MeMViT-B | 50% (uni) | 2304 | 37.71 |
| MeMViT-B | 100% | 4608 | 37.87 |
| **AMViT-B** | 50% (uni) | 1200 | 38.06 |
| **AMViT-B** | 100% | 1800 | **38.45** |

## 5    CONCLUSION

The optimal temporal receptive field (TRF) is essential in achieving long-form video understanding using Vision Transformer (ViT) models. Although memory-augmented methods could extend TRF by caching the stop-gradient Key-Value (KV) pairs, they still have a fixed TRF and the first-in-first-out caching strategy could lead to loss of salient information. To tackle the challenges posed by varying duration and diverse dynamics in long-form videos, we propose the Adaptive Memory Mechanism (AMM), which allows ViT models to adaptively adjust the TRF based on the input video. AMM features an input-aware selective module that selects tokens from previous iterations that are relevant to the input in the subsequent iteration. By using a memory bank, AMM retains essential information that exceeds the memory length limit. Experimental results show that the proposed method outperforms the state-of-the-art methods in action detection, recognition, and anticipation tasks.

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

# A APPENDIX

## A.1 ALGORITHM

---

**Algorithm 1** Tokens Selection Function (torch style)

---

"""

$q$ is the Query of Class Token;

$cached\_k$ and $cached\_v$ are the Key and Query filtered by $q$;

$len$ is the number of selected tokens;

"""

1: **def** **kv_selection**( $q$, $cached\_k$, $cached\_v$, $len$ ) :

     # Batch Size (b), Head Number (h), Number of Tokens (n) and Dimension of Single Head (d)

2:     $b$, $h$, $\_$, $d = q.shape$

     # @ means matrix multiplication

3:     $atten\_score = (\, q[:, :, :1]\, @\, cached\_k.transpose(-2, -1)).reshape(b,\, h,\, -1\,)$

     # Get len top tokens' indices based on Attention Score

4:     $\_,\, indices = torch.topk(\, atten\_score,\, len,\, dim = 2\,)$

5:     $indices = indices.unsqueeze(-1).expand(-1, -1, -1, d)$

     # Select out the top tokens;

6:     $selected\_k = torch.gather(cached\_k,\, 2,\, indices)$

7:     $selected\_v = torch.gather(cached\_v,\, 2,\, indices)$

8:     **return** $selected\_k$, $selected\_k$

---

---

**Algorithm 2** AMViT Attention (torch style)

---

1: **class** **AMAttention**( ) :

     # compressed KV cached

2:     $self.cached\_k = [\,]$

3:     $self.cached\_v = [\,]$

     # Memory bank

4:     $self.bank\_k = None$

5:     $self.bank\_v = None$

6:     **def** **forward**$(q, k, v)$

     # select out class token to calculate attention score with KV cached and Memory bank

7:     $cls = q[\,:, :1, :\,]$

     # using kv_selection function get selected K, V and bank K, V.

8:     $selected\_k, selected\_v = kv\_selection(cls, self.cached\_k, self.cached\_v, len)$

9:     $self.bank\_k, self.bank\_v = kv\_selection(cls, self.bank\_k, self.bank\_v, len)$

     # concatenation along token dimension

10:    $k\_a = torch.cat(self.bank\_k, selected\_k, k, dim = 2)$

11:    $v\_a = torch.cat(self.bank\_v, selected\_v, v, dim = 2)$

12:    $x = attn(q, k\_a, v\_a)$

13:    **return** $x$

---

## A.2 IMPLEMENTATION DETAILS

Table 5: ViT-B Hyper Parameters in different datasets

| config | AVA | Kitchen Recognition | Kitchen Anticipation |
|---|---|---|---|
| optimizer | | AdamW | |
| optimizer momentum | | $\beta_1, \beta_2 = 0.9, 0.999$ | |
| weight decay | | 0.05 | |
| learning rate schedule | | cosine decay | |
| learning rate | 2.5e-4 | 5e-4 | 5e-4 |
| batch size | | 64 | |
| warmup epochs | | 5 | |
| total epochs | | 30 | |
| input frame | | 8 | |
| drop path | | 0.2 | |

Table 6: Model Setting in ViT-B

| config | MemViT-B | MemViT-L | AMViT-B | AMViT-L |
|---|---|---|---|---|
| memory bank | * | * | 50 | 50 |
| # of kv cache | | 2 | | |
| # of cpr(kv) | | 192 | | |
| selection() | * | * | 50 | 50 |
| ratio($\alpha$) | * | * | 0.2 | 0.2 |

