# OpenReview forum: "Adaptive Memory Mechanism in Vision Transformer for Long-form Video Understanding"
_ICLR.cc/2025/Conference — Submitted to ICLR 2025_

### Official Review · Reviewer_j6eW · 2024-10-23

**Soundness:** 3
**Presentation:** 2
**Contribution:** 3
**Rating:** 5
**Confidence:** 3

**Summary:**

This paper introduces an Adaptive Memory Mechanism (AMM) to improve Vision Transformers (ViT) for long-form video understanding. AMM dynamically adjusts the Temporal Receptive Field (TRF) based on video content, overcoming limitations of fixed TRF approaches that either lose key information or increase computational costs. Experiments show that AMViT, integrating AMM, outperforms existing models like MeMViT in tasks such as action recognition, anticipation, and detection, while reducing computational overhead, validated on datasets like AVA and Epic-Kitchens.

**Strengths:**

1. Long-form video understanding is an important task, and efficiency is indeed a crucial metric in this context.

2. The proposed method can reduce both training and inference costs.

3. Introducing a memory bank to handle long sequence inputs is intuitive and reasonable.

**Weaknesses:**

1. (important) The number of benchmarks (only 2) and baselines (also only 2) compared seems somewhat limited. Adding more experiments would make the paper more convincing.

2. (important) Although the authors emphasize that the new architecture is designed for long-form video, this aspect is not discussed in the experimental section. Are the benchmarks presented in the paper truly for long videos, and what is the average input length? It would have been better if the authors had conducted more detailed evaluations on benchmarks like MovieChat-1K [1] or LongVideoBench [2].

3. The writing and figures in the paper need improvement, especially regarding the notation for memory. There are too many subscripts and superscripts, along with the extensive use of qkv notations, which made it take me three times longer to understand the entire paper.

[1] Song, Enxin, et al. "Moviechat: From dense token to sparse memory for long video understanding." Proceedings of the IEEE/CVF Conference on Computer Vision and Pattern Recognition. 2024.

[2] Wu, Haoning, et al. "Longvideobench: A benchmark for long-context interleaved video-language understanding." arXiv preprint arXiv:2407.15754 (2024).

**Questions:**

Please revise the Weaknesses section point by point. This is a paper with great potential. If the authors can provide additional responses to certain issues, discuss related work more thoroughly, and include more experiments and observations, I would be very happy to raise my score.

---

> ### Author Response · Authors · 2024-11-25
> **Answer for Reviewer j6eW**
>
> Thanks for the suggestions.
>
> **Limited Benchmarks and Baselines:** As suggested, we extended our performance comparison by including a number of additional SoTA methods. As tabulated in Tables 1-4 in the supplementary materials for the extra experiments comments, the proposed AMViT outperforms all the SoTA methods, which further demonstrates the effectiveness of AMViT. In particular, we included another benchmark, Diving48, which is a comprehensive dataset specifically designed for fine-grained action recognition in diving videos. This dataset provides a diverse range of diving actions and is widely used to evaluate the performance of models in capturing intricate motion details.
>
> **Average Input Duration:** The benchmarks we used are long videos, where the average input duration is comparable to or longer than one of the suggested benchmark LongVideoBench.
> | Dataset         | Average Duration (s) |
> | ------------- |:-----------------:|
> | LONGVIDEOBENCH       | 473             |
> | AVA      | 900             |
> | Epic-kitchen-100     |  6120            |
> | Diving48     |  378            |
>
> We will elaborate this point in the experiment section as suggested.
>
> Writing and Figures: We agree the presentation clarity of the paper should be further enhanced.

---

> > ### Comment · Reviewer_j6eW · 2024-11-25
> >
> > Thanks for your response. I've take a look at other reviewer's comments and also MemViT paper again. It seems to me AMViT lack a part of efficiency analysis, which is the important part MemViT wanna present (for example fig1 and 3 in MemViT paper). And the paper presentation also much stronger than proposed AMViT. Therefore, I decide to keep my rating 5 now.

---

### Official Review · Reviewer_a8AG · 2024-11-03

**Soundness:** 2
**Presentation:** 3
**Contribution:** 2
**Rating:** 3
**Confidence:** 4

**Summary:**

The paper proposes an Adaptive Memory Mechanism (AMM) for Vision Transformer (ViT) in long-form video understanding. It addresses the issue of selecting an optimal Temporal Receptive Field by allowing ViT to adjust TRF dynamically. Instead of directly discarding early Key-Value cache, AMM uses a Memory Bank to retain important embeddings from the Key-Value cache based on attention scores. Experiments on AVA and Epic-Kitchens show the advantages of AMM in action recognition, anticipation, and detection tasks.

**Strengths:**

1.Long-form video understanding is an important research task, and the author has provided a reasonable solution.

2.The paper is well-written, making it easy to read.

**Weaknesses:**

1.The novelty of memory bank is limited. Many studies have explored how to utilize memory to retain important historical information and how to dynamically update memory. For example, Xmem[1] prioritizes retaining the most frequently used candidates. MA-LLM[2] and MovieChat[3] merge the two most similar candidates based on similarity once the memory bank capacity is exceeded. The innovations and advantages of the memory bank proposed in this paper compared to these methods are unclear.

2.The fairness of the experiment is in question. When comparing with the baseline model MeMViT, the authors replaced the backbone of MeMViT from MViT to UMT. This seems to have led to a decline in the performance of the baseline model. For example, in the EPIC-KITCHEN-100 action recognition task, the performance reported in the original paper on MeMViT was 48.4%, while the performance presented in this paper is 43.03%. The authors should maintain the same settings as MeMViT for the experiments to make the results more credible.

3.The performance improvement is limited. Compared to the baseline model MeMViT, the performance improvement is less than 1% in all experiments.

4.Lacks of comparison with the latest methods. This article only presents comparisons with ViT and MeMViT. Some recent methods are missing, such as MAT[4] and MC-ViT[5].

5.Lacks of necessary ablation studies. (2) This paper uses an input-aware selective module to prevent redundant embeddings from being retained, and uses a memory bank to retain useful embeddings. However, there are no ablation experiments to demonstrate the effectiveness of these two components individually. (2) The lack of ablation experiments on the memory bank update method. For example, comparing the update of the memory bank using attention score of class tokens proposed in this paper with previous methods (see weakness 1) and First-In-First-Out (FIFO).

[1] XMem: Long-Term Video Object Segmentation with an Atkinson-Shiffrin Memory Model, ECCV 2022
[2] MA-LMM: Memory-Augmented Large Multimodal Model for Long-Term Video Understanding, CVPR 2024
[3] MovieChat: From Dense Token to Sparse Memory for Long Video Understanding, CVPR2024
[4] Memory-and-Anticipation Transformer for Online Action Understanding, ICCV 2023
[5] Memory Consolidation Enables Long-Context Video Understanding, arxiv 2024

**Questions:**

When comparing with MeMViT, your model uses the memory bank and the selected Q-V cache, while MeMViT only uses Q-V cache. Have you ensured that the number of embeddings in both model is consistent? Specifically, does the size of the memory bank plus the size of the selected Q-V cache match the size of the unselected Q-V cache?

---

> ### Author Response · Authors · 2024-11-25
> **Answer for Reviewer a8AG**
>
> Thanks for the comments.
>
> **Comparing AMViT with Xmen [15], MC-ViT [13], MA-LLM [16], MovieChat [17] and MAT [5]:**
> XMem [2] uses the memory mechanism at the image level for video object segmentation. Though the high-level idea is the same, the proposed AMViT proposes a novel memory mechanism at the image-patch level, and the application is for action recognition and detection, where one of the crucial steps is to identify the relevant salient features.
>
> MC-ViT [3] uses a memory consolidation mechanism to capture longer context. The proposed AMViT utilizes the proposed input-aware selection mechanism so that more relevant context over time can be more effectively captured For fair comparisons, we evaluated MC-ViT using the same backbone of AMViT and reported the result in Table 4. It is noted that both MemViT and AMViT can achieve better performance than MC-ViT.
>
> MA-LLM [4] & MovieChat [5] make use of memory compression based on the adjacent similarity to implement long-term memory. AMViT instead makes use of an input-aware selection mechanism to adaptively maintain the memory bank.
>
> MAT [6] uses a memory-anticipation-based paradigm to model the temporal structure of past, present, and future. AMViT uses the adaptive memory mechanism instead. We also carried out additional performance comparisons, as shown in Table 1. We find that AMViT can outperform MAT for the action anticipation task based on the Epic-kitchen-100 dataset.
>
> **About Reproduced Results for MeMViT:** MeMViT does not provide the code for training the model on Epic-kitchen. We utilize the code from RULSTM [1], which may be the reason for the disparity. For the AVA dataset, our reproduced results are consistent with those reported in the original paper. Despite this, our reproduced results remain SoTA according to their evaluation methodology. Detailed information can be found in Tables 1 and 2 of the supplementary experiments comment.
>
> In addition, we also tried to reproduce the results of MeMViT by adopting the backbone adopted in the original paper.
> | | AVA (action detection) mAP(%) | Epic-kitchen (action recognition) top-1 acc(%) | Epic-kitchen (action anticipation) top-1 acc(%) |
> |---------|-----------|-------------------|---------------------------|
> | Results in original paper | 29.3 | 46.2| 15.1 |
> | Our reproduced results| 29.17| 38.45 | 19.5|
>
> **Marginal Improvement:** Although our method does not show a very significant improvement, it achieves better results requiring less augmented KV pairs. Furthermore, we tested our method on another benchmark, Diving48, where it achieved a 1.4% improvement over MeMViT. For more details, please refer to Table 4 in the supplementary experiments section.
>
> **Latest Methods Comparisons:** As suggested, we extended our performance comparison by including a number of additional SoTA methods. As tabulated in Tables 1-4 in the supplementary materials for the extra experiments comments, the proposed AMViT outperforms all the SoTA methods, which further demonstrates the effectiveness of AMViT.
>
> **Ablations in ISM and Memory Bank:** We added the suggested ablation study. According to Table 5 in Additional Experimental Results, we show that adding the ISM can enhance the performance. Adding also the Memory Bank can further enhance the performance. AMViT without ISM and Memory Bank degenerates back to MeMViT. Note that even though AMViT needs an extra Memory Bank to store the important compressed KV cache pairs, the total number of augmented tokens is reduced because the ISM module further shrinks the compressed KV cache pairs via the selection.
>
> **KV Cache Consistency:**  In our experiments, we kept all modules (compressed KV cache, backbone, etc.) the same, with the only differences being the addition of the ISM and the memory bank in AMViT. Please see Table 5 in the additional experimental results.
>
> **References:**
> [1] What Would You Expect? Anticipating Egocentric Actions With Rolling-Unrolling LSTMs and Modality Attention. [ICCV’2019]
> [2] XMem: Long-Term Video Object Segmentation with an Atkinson-Shiffrin Memory Model. [ECCV’2022]
> [3] Memory Consolidation Enables Long-Context Video Understanding. [ICML’2024]
> [4] MA-LMM: Memory-Augmented Large Multimodal Model for Long-Term Video Understanding. [CVPR’2024]
> [5] MovieChat: From Dense Token to Sparse Memory for Long Video Understanding. [CVPR’2024]
> [6] Memory-and-Anticipation Transformer for Online Action Understanding. [ICCV’2023]

---

### Official Review · Reviewer_CdCg · 2024-11-04

**Soundness:** 2
**Presentation:** 2
**Contribution:** 2
**Rating:** 5
**Confidence:** 5

**Summary:**

This paper aims to enhance ViT for long-term video understanding. The authors design a memory bank to store historical information and develop input-aware adaptive memory selection to retrieve the relevant information to assist long-term analysis. The experiments show that the architecture demonstrates satisfactory performance with high efficiency.

**Strengths:**

1. The analysis of the limited temporal receptive field in long-term video understanding makes sense, and the motivation is clear.
2. The method is simple and intuitive.

**Weaknesses:**

1. The experiments are limited. Only AVA and Epic-Kitchens are reported. Results on more video datasets are required to verify the effectiveness of the adaptive memory design. Besides, the performance improvements are marginal.
2. The memory bank is recurrently updated by adaptive selection. Is it possible that in a long video, the content in the middle of the video is not closely related to the beginning, and only relevant content appears towards the end? However, during the memory bank update process, the tokens of the earlier video content were already discarded.

**Questions:**

1. Does the KV Cache in this paper retain the gradient?
2. This paper focuses on a pure vision model with enhanced memory design. However, the ViT-only architecture is capable of a limited range of video-related tasks. Is it possible to integrate it with video-language models to achieve wider range of video tasks to exert more impact on the community?

---

> ### Author Response · Authors · 2024-11-25
> **Answer for Reviewer CdCg**
>
> Thanks for the comments.
>
> **State-of-the-Art Comparisons:** As suggested, we extended our performance comparison by including a number of additional SoTA methods. As tabulated in Tables 1-4 in the supplementary materials for the extra experiments comments, the proposed AMViT outperforms all the SoTA methods, which further demonstrates the effectiveness of AMViT.
>
> **Marginal Improvement:** Although our method does not show a very significant improvement, it achieves better results requiring less augmented KV pairs. Furthermore, we tested our method on another benchmark, Diving48, where it achieved a 1.4% improvement over MeMViT. For more details, please refer to Table 4 in the supplementary experiments section.
>
> **Limitation of Adaptive Selection:** It is true that if the range of dependency is long, the SOTA memory-augmented methods, such as SWIM [1], MeMViT [2], and MC-ViT [3], as well as the proposed AMViT may have the irrelevant content discarded. While AMViT cannot eliminate the possibility, it tries to alleviate this limitation by maintaining an adaptive temporal receptive field. The situations mentioned by the reviewer could indeed happen, and more robust memory mechanisms which can effectively handle that are worth further investigation.
>
> **Gradient:** The KV Cache does not retain the gradient.
>
> **Vision-language Extension:** Our methods, being memory-augmented, can potentially be integrated into language models. MC-ViT [3] has already been explored to demonstrate the possibility of integration of augmented methods with language models, indicating the feasibility of such an extension. This will be our future work.
>
> **References:**
> [1] Video Swin Transformer. [CVPR’2022]
> [2] MeMViT: Memory-Augmented Multiscale Vision Transformer for Efficient Long-Term Video Recognition. [CVPR’2022]
> [3] Memory Consolidation Enables Long-Context Video Understanding. [ICML’2024]

---

### Official Review · Reviewer_TRqz · 2024-11-04

**Soundness:** 1
**Presentation:** 2
**Contribution:** 1
**Rating:** 5
**Confidence:** 4

**Summary:**

This paper presents an adaptive memory method to improve the existing memory-augmented methods for long-form video understanding. The method is based on MeMViT but makes the memory bank adaptive to support the adaptive temporal receptive field. The experiments are conducted on Ava and Epic-Kitchens dataset with the comparison with ViT and MeMViT.

**Strengths:**

* Long-form video understanding is an important video research topic and the idea of using an adaptive memory bank sounds reasonable and promising.
* Compared to MeMViT, the results show consistent improvements though some datasets only have marginal gain.

**Weaknesses:**

* One of the main motivations of the paper is to retain embeddings instead of discarding memory when the memory limit is reached. However, based on the experiments, it's unclear if the effective receptive field of AMViT is indeed larger than MeMViT through the proposed adaptive memory module. Are they still using the same memory bank size?
* In the model section, the paper presents two new modules, including Input-aware selective module (ISM) and Adaptive Memory mechanism(AMM). However, there are no ablations to validate the individual effectiveness of these modules.
* How do we select parameters for MeMViT? Some parameters for MeMViT (Table 6) are not defined, e.g, memory bank size. Is it the same as AMViT? Given the authors are reproducing MeMViT with a different backbone, how the results compare to the original paper.
* In Table 1, it's unclear why all the three methods are having the same FLOPs and parameters given MeMviT and AMViT has additional memory bank modules. It's also better to conduct run-time comparison.
* The experiments are also missing a system-level comparison with the current SOTA results on the benchmarks.

**Questions:**

Please see weaknesses

---

> ### Author Response · Authors · 2024-11-25
> **Answer for Reviewer TRqz**
>
> Thank you for the comments.
>
> **Memory Bank Size of MeMViT and AMViT:**  The proposed AMViT introduces an Input-aware Selective Module (ISM) to retain the important compressed KV cache and an additional Memory Bank to store the cache that would otherwise be dropped in MeMViT. Therefore, there is no Memory Bank in MeMViT. Only the KV cache of the video segment immediately preceding the current segment is considered in MeMViT. For fair comparisons, we kept all modules (compressed KV cache, backbone, etc.) the same in our experiments for both MeMViT and AMViT, with the only differences being the addition of the ISM and the memory bank in AMViT.
>
> **Ablations in ISM and Memory Bank:** We added the suggested ablation study. According to Table 5 under the supplementary experiments comments, AMViT without ISM and Memory Bank noticeably would degenerate to MeMViT. Even though AMViT needs an extra Memory Bank to store the important compressed KV cache, the total number of augmented tokens is reduced because the ISM module further shrinks the compressed KV cache via the selection.
>
> **Effective Temporal Receptive Field:** Please check Figure 4 in the paper, it shows that the memory bank keeps more tokens near recent iterations for the "Take Pot" action than for the "Wash Pot" action. This suggests the Memory Bank focuses more on recent iterations for "Take Pot." This aligns with Figure 1, which indicates the pot appears for a shorter duration during "Take Pot" than "Wash Pot." The different retained tokens based on different actions indicate that AMViT would have a larger temporal receptive field than MeMViT when it is needed for particular actions.
>
> **Parameter Setting and MeMViT Performance based on Original Backbone:** As explained, there is no Memory Bank in MeMViT. The sign “*” in Table 6 is to indicate that. We will add a footnote accordingly. Also, as suggested, we tried to reproduce the results of MeMViT using the backbone adopted in the original paper.
>
> | | AVA (action detection) mAP(%) | Epic-kitchen (action recognition) top-1 acc(%) | Epic-kitchen (action anticipation) top-1 acc(%) |
> |------------|-------------|------------|------------------|
> | Results in original paper | 29.3 | 46.2| 15.1 |
> | Our reproduced results| 29.17| 38.45 | 19.5|
>
> For the AVA dataset, our reproduced results are consistent, while there is a significant discrepancy in the Epic-kitchen dataset. As MeMViT does not provide the code for training the model on Epic-kitchen, we utilize the code from RULSTM [1], which may be the reason for the disparity. Despite this, our reproduced results remain SoTA according to their evaluation methodology. Details can be found in Tables 1 and 2 under the supplementary experiments comment.
>
> **FLOPs and # Parameters:** (i) We apologize for the typos in Table 1 of the submitted manuscript, which should be the same as in Tables 2 and 3. (ii) With reference to the ViT backbone, the additional FLOPs and the number of parameters due to the memory-augmented methods are just incremental and insignificant, which corresponds to the additional CNN for compressing the KV cache. AMViT does not intend to reduce the model size but tries to make more effective use of the augmented memory to achieve dynamic temporal receptive fields, and thus better performance.
>
> **State-of-the-Art Comparisons:** As suggested, we extended our performance comparison by including a number of additional SoTA methods. As tabulated in Tables 1-4 of the additional experimental results we provided, the proposed AMViT outperforms all the SoTA methods, which further demonstrates the effectiveness of AMViT.
>
> **References:**
> [1] What Would You Expect? Anticipating Egocentric Actions With Rolling-Unrolling LSTMs and Modality Attention. [ICCV’2019]

---

> > ### Comment · Reviewer_TRqz · 2024-12-03
> >
> > Thank the authors for their response. After reviewing the rebuttal and the feedback from other reviewers, I feel some of my concerns have been addressed. However, the improvements over the MeMViT baseline remain limited in both novelty and performance gains. Based on this, I am adjusting my rating to a 5, but I still think it falls below the acceptance threshold.

---

### Official Review · Reviewer_gHkC · 2024-11-04

**Soundness:** 2
**Presentation:** 2
**Contribution:** 2
**Rating:** 5
**Confidence:** 4

**Summary:**

The paper addresses a solution for better long-form video understanding using a method named Adaptive Memory Mechanism (AMM). This method enables the Vision Transformer (ViT) to adjust its temporal receptive field dynamically depending on the input video. A memory bank is utilized to save the most important Key-Value when temporally processing the videos. The proposed method is tested on AVA and Epic-Kitchens datasets for action detection, recognition, and anticipation tasks. Experiment results show performance improvement to the ViT baselines without additional cost.

**Strengths:**

1. The method have better performance than baselines without additional cost.

**Weaknesses:**

1. The paper lacks SoTA comparisons. Is the task different from common action recognition and action detection? Multiple methods such as VideoMAE, Omnivore, or MMT have been tested on these datasets. It would be helpful if the authors could explain the difference between previous SoTAs with the proposed method, for example in parameter count or GFLOP difference.
2. The improvement to ViT and MeMVit baselines is marginal.
3. There is no difference in the FLOPs and Param(M) numbers compared to the baselines. Can the authors explain further the efficiency advantage achieved by the proposed method?

**Questions:**

1. Will there be a significant performance difference if the model is not pre-trained with UMT?

---

> ### Author Response · Authors · 2024-11-25
> **Answer for Reviewer gHkC**
>
> Thanks for the suggestions.
>
> **State-of-the-Art Comparison:** As suggested, we extended our performance comparison by including a number of additional SoTA methods. As tabulated in Tables 1-4 of the additional experimental results we provided, the proposed AMViT outperforms all the SoTA methods, which further demonstrates the effectiveness of AMViT.
>
> **Action Recognition and Action Detection Tasks:** We test on the same Action Recognition and Action Detection tasks as in other related works like VideoMAE and Omnivore. Yet, the way how the proposed AMViT trains the ViT is different from methods like VideoMAE and Omnivore which process a limited number of frames at a time and thus focus on the action occurrence duration within the video. In contrast, our method processes the video stream continuously fed to the model, starting from the beginning, to perform action recognition or detection tasks. We will clarify this point in the paper.
>
> **AMViT vs. “VideoMAE, Omnivore, UMT”:** Methods like VideoMAE, Omnivore, and UMT are designed for understanding short-form videos, and they cannot scale if directly applied to long-from videos. The quadratic increase in complexity will be resulted as the length of the video segment increases due to the use of the Transformer architecture. Memory-augmented methods like MeMViT are introduced to address the issue by augmenting compressed versions of the embeddings of the preceding segments. The proposed AMViT further extends the idea to capture longer-range dependency with an adaptive memory augmentation mechanism. It is to be noted that the advantage gained by the proposed adaptive memory mechanism is orthogonal to how we train the ViT like VideoMAE, Omnivore, and UMT.
>
> **Marginal Improvement:** Although our method does not show a very significant improvement, it achieves better results while reducing the number of augmented KV pairs. Furthermore, we tested our method on another benchmark, Diving48, where it achieved a 1.4% improvement over MeMViT. For more details, please refer to Table 4 in the supplementary experiments section.
>
>
> **FLOPs and # Parameters:** With reference to the ViT backbone, the additional FLOPs and the number of parameters due to the memory-augmented methods are often insignificant, mainly due to the additional CNN required to compress the KV cache. Yet AMViT can make more effective use of the augmented memory to achieve dynamic temporal receptive fields, and thus better performance. Note: There are typos in Table 1 of the submitted manuscript. The FLOPs and the number of parameters should be:
> FLOPs(M): ViT-B(202.22), MeMViT(202.28), AMViT(202.28) &
> Parameters(M): ViT(85.64), MeMViT(85.66), AMViT(85.66).
>
> **Using Other Pre-trained Models:** While the choice of the pre-trained model shall affect the overall performance, the contribution of AMViT is how to get the most from a particular pre-trained model by achieving adaptive temporal receptive fields. Different backbones, due to how they are trained, should have different performance. In principle, applying the proposed adaptive memory mechanism can further improve their performance.

---

> > ### Comment · Reviewer_gHkC · 2024-12-02
> >
> > Thank you for clarifying the SoTA comparison and the difference between the proposed method and other related works. I highly appreciate the effort of adding the SoTA and additional ablation studies performed by the authors to address the reviewers' concerns, however, I think the paper still lacks a significant advantage over the closely related MeM-ViT. Nevertheless, I decided to change my score to 5.

---

### Author Response · Authors · 2024-11-25
**SoTA Comparison**

As suggested by all the reviewers, we have added back the SOTA methods suggested by the reviewers and those we further identified to provide a more comprehensive performance comparison. Most of them are the results reported in their original papers, and some are reproduced by us for fair comparison. (* represent the results that are reproduced by ourselves.)

Table 1: Performance Comparison. (Dataset: Epic-kitchen-100; Evaluation Task: Action Anticipation)
| Model         | Pre-train Dataset |Top-1 Acc (%)  |
| ------------- |:-----------------:|:-------------:|
| RU-LSTM [1]   | IN-1K             |13.3           |
| AVT [2]       | IN-1K             |13.6           |
| DCR [3]       | IN-1K             |14.6           |
| TeSTra[4]     | IN-1K             |17.0           |
| MAT [5]       | IN-1K             |18.8           |
| ViT-B*         | K710              |19.3           |
| MeMViT-B*      | K710              |19.5           |
| AMViT-B(Our)  | K710              |19.8           |
| AMViT-L(Our)  | K710              |22.6           |

Table 2: Performance Comparison. (Dataset: Epic-kitchen-100; Evaluation Task: Action Recognition)
| Model         | Pre-train Dataset |Top-1 Acc (%)  |
| ------------- |:-----------------:|:-------------:|
| TSN [6]       | IN-1K             |20.54          |
| LSTA [7]      | IN-1K             |30.33          |
| VNMCE [8]     | IN-1K             |29.00          |
| RU-LSTM [1]   | IN-1K             |33.06          |
| ViT-B*         | K710              |37.12          |
| MeMViT-B*      | K710              |37.87          |
| AMViT-B(Our)  | K710              |38.60          |
| AMViT-L(Our)  | K710              |43.15          |

Table 3: Performance Comparison. (Dataset: AVA; Evaluation Task: Action Detection)
| Model         | Pre-train Dataset |Param(M)|mAP (%)  |
| ------------- |:-----------------:|:------:|:------------:|
| SlowFast[9]   | K600              |59      |27.5          |
| X3D-XL [10]   | K600              |11      |27.4          |
| MViT [11]     | K700              |51      |31.8          |
| ViT-B*         | K710              |86      |28.59         |
| MeMViT-B*      | K710              |86      |29.17         |
| AMViT-B(Our)  | K710              |86      |30.07         |
| AMViT-L(Our)  | K710              | 307    |35.98         |

Table 4: Performance Comparison. (Dataset: Diving48 [14]; Evaluation Task: Action Recognition)
| Model         |Param(M)|Top-1 Acc (%))  |
| ------------- |:------:|:------------:|
|TimeSformer [12]|121    |74.9          |
| MC-ViT [11]*    |86     |74.5          |
| ViT-B*         |86      |75.1         |
| MeMViT-B*      |86      |76.2         |
| AMViT-B(Our)  |86      |77.6         |

**Reference:**
[1] What Would You Expect? Anticipating Egocentric Actions with Rolling-Unrolling LSTMs and Modality Attention. [ICCV’2019]
[2] Anticipative Video Transformer. [ICCV’2021]
[3] Learning to Anticipate Future with Dynamic Context Removal. [CVPR’2022]
[4] Real-time Online Video Detection with Temporal Smoothing Transformers. [ECCV’2022]
[5] Memory-and-Anticipation Transformer for Online Action Understanding. [ICCV’2023]
[6] Temporal Segment Networks for Action Recognition in Videos. [TPAMI’2019]
[7] LSTA: Long Short-Term Attention for Egocentric Action Recognition. [CVPR’2019]
[8] Leveraging Uncertainty to Rethink Loss Functions and Evaluation Measures for Egocentric Action Anticipation. [ECCV’2018]
[9] SlowFast Networks for Video Recognition. [ICCV’2019]
[10] X3D: Expanding Architectures for Efficient Video Recognition. [CVPR’2019]
[11] Improved Multiscale Vision Transformers for Classification and Detection. [CVPR’2022]
[12] Is Space-time Attention All You Need for Video Understanding? [ICML’2021]
[13] Memory Consolidation Enables Long-Context Video Understanding. [ICML’2024]
[14] RESOUND: Towards Action Recognition without Representation Bias. [ECCV’2018]

---

> ### Comment · Reviewer_TRqz · 2024-12-03
> **Which rows are reproduced by authors?**
>
> * Although the authors mentioned "* represent the results that are reproduced by ourselves", it seems no rows have this mark?

---

> ### Author Response · Authors · 2024-12-03
>
> Sorry for the missing annotation, we have updated it.

---

### Author Response · Authors · 2024-11-25
**Module Ablation**

Table 5 Ablation study based on the Epic-Kitchen dataset and the evaluation task of action recognition. In the experiment, a 12-layer ViT-B is adopted for all the methods. Settings for each transformer are reported. Note that the number of compressed KV pairs obtained via CNN per video segment is 197. The accuracy reached the maximum when the selection size is 60 where the total number of augmented compressed KV pairs is 170 as compared to 394 for MeMViT.
| Model              | Memory length (m) | Total # of compressed <br> KV pairs (m*197) | Memory bank size (L) | Selection size (k) |Total # of selected <br> KV pairs (m*k)|Total # of augmented <br> KV pairs (L+m*k) | Acc.   |
|---------|------------|--------------|-------------|--------------|-------------|--------------------|--------|
| ViT-B              | 0            | N/A               | N/A            | N/A          | N/A                   |N/A           | 37.12  |
| MeMViT-B       | 1         | 197               | N/A        | N/A          | 197         |197       | 37.96  |
| MeMViT-B        | 2       | 394        | N/A            | N/A         | 394           |394         | 37.87  |
| AMViT-B (no Memory Bank) | 2    | 394           | 0         | 50                | 100        |100          | 38.12  |
| AMViT-B        | 1             | 197              | 50         | 50         | 50         |100         | 38.18  |
| AMViT-B            | 2         | 394              | 50        | 30         | 60         |110         | 38.43  |
| AMViT-B            | 2         | 394     | 50        | 50       | 100         |150            | 38.45  |
| AMViT-B            | 2        | 394    | 50         | 60         | 120         |170         | 38.60  |
| AMViT-B            | 2           | 394        | 50      | 70          | 140         |190          | 38.51  |

---

> ### Comment · Reviewer_TRqz · 2024-12-03
> **Complexity comparison**
>
> For different variants here, it might be better to also include a complexity or run-time comparison.

---

### Meta-Review · Area_Chair_dQNk · 2024-12-18

**Metareview:**

(a) Scientific Claims and Findings

The paper introduces an Adaptive Memory Mechanism (AMM) for Vision Transformers (ViT) aimed at improving long-form video understanding. AMM allows ViT to dynamically adjust its Temporal Receptive Field (TRF) based on video content, utilizing a memory bank to retain important Key-Value pairs. The method is evaluated on AVA and Epic-Kitchens datasets, demonstrating performance improvements over ViT baselines without additional computational cost.

(b) Strengths

Reviewer gHkC highlights that the method improves performance over baselines without extra computational cost. TRqz and j6eW note that the approach is relevant to the important task of long-form video understanding and is efficient in reducing both training and inference costs. CdCg and a8AG appreciate the clarity and readability of the paper, with CdCg also noting the simplicity and intuitiveness of the method.

(c) Weaknesses

A major concern, as pointed out by reviewers gHkC, TRqz, and a8AG, is the lack of comparisons with state-of-the-art methods, which raises questions about the method's relative performance. TRqz and CdCg also mention that the performance improvements are marginal. The experiments are limited to two datasets, as noted by CdCg and j6eW, and the paper lacks ablation studies to validate the contributions of individual modules, as highlighted by TRqz and a8AG. Additionally, a8AG questions the novelty of the memory bank, and there are concerns about the fairness of the experimental setup, particularly in comparisons with MeMViT.

(d) Decision Reasons

The AC aligns with the reject recommendation of all reviewers. The decision to reject the paper is primarily due to the lack of comprehensive comparisons with state-of-the-art methods and limited experimental validation, as noted by reviewers gHkC, TRqz, and j6eW. The marginal performance gains over existing methods, as mentioned by TRqz and CdCg, do not provide sufficient justification for acceptance. Furthermore, the paper's contributions are not deemed novel or significant enough compared to existing work, as pointed out by a8AG. Overall, while the paper addresses an important problem, it requires more robust experiments, comparisons, and analyses to strengthen its contributions.

**Additional Comments On Reviewer Discussion:**

During the rebuttal period, the authors addressed several concerns raised by the reviewers, leading to some adjustments in their evaluations, though not enough to change the overall decision.

Reviewer gHkC appreciated the authors' efforts in clarifying the state-of-the-art comparisons and adding additional ablation studies. However, they felt that the paper still lacked a significant advantage over the closely related MeMViT. Despite these improvements, the reviewer decided to adjust their score to 5, indicating a marginal improvement in their perception of the paper.

Reviewer TRqz acknowledged that some of their concerns were addressed in the rebuttal, particularly after considering feedback from other reviewers. However, they maintained that the improvements over the MeMViT baseline were limited in both novelty and performance gains. Consequently, they adjusted their rating to a 5, but still considered the paper to fall below the acceptance threshold.

Reviewer j6eW noted that the rebuttal did not sufficiently address the lack of efficiency analysis, which was a key aspect of the MeMViT paper. They also found the presentation of the AMViT paper to be weaker compared to MeMViT. As a result, they decided to keep their rating at 5.

In weighing these points for the final decision, the primary considerations were the limited novelty and marginal performance improvements over existing methods, as well as the lack of comprehensive efficiency analysis. Despite the authors' efforts to address the reviewers' concerns, the paper did not convince them towards accepting the paper.

---

### Decision · Program_Chairs · 2025-01-22

Reject